# The Opportunities and Challenges of Mesenchymal Stem Cells-Derived Exosomes in Theranostics and Regenerative Medicine

**DOI:** 10.3390/cells13231956

**Published:** 2024-11-25

**Authors:** Sachin Yadav, Pritiprasanna Maity, Kausik Kapat

**Affiliations:** 1Department of Medical Devices, National Institute of Pharmaceutical Education and Research Kolkata, 168, Maniktala Main Road, Kankurgachi, Kolkata 700054, West Bengal, India; sachiny899@gmail.com; 2School of Medicine, University of California Riverside, Riverside, CA 92525, USA

**Keywords:** mesenchymal stem cells, exosome, therapeutic, theranostic, tissue regeneration, wound healing, cancer

## Abstract

Cell-secreted nanovesicles of endosomal origin, called exosomes, are vital for mediating intracellular communication. As local or distal transporters of intracellular cargo, they reflect the unique characteristics of secretory cells and establish cell-specific interactions via characteristic surface proteins and receptors. With the advent of rapid isolation, purification, and identification techniques, exosomes have become an attractive choice for disease diagnosis (exosomal content as biomarkers), cell-free therapy, and tissue regeneration. Mesenchymal stem cell (MSC)-derived exosomes (MSC-exosomes) display angiogenic, immune-modulatory, and other therapeutic effects crucial for cytoprotection, ischemic wound repair, myocardial regeneration, etc. The primary focus of this review is to highlight the widespread application of MSC-exosomes in therapeutics, theranostics, and tissue regeneration. After a brief introduction of exosome properties, biogenesis, isolation, and functions, recent studies on therapeutic and regenerative applications of MSC-exosomes are described, focusing on bone, cartilage, periodontal, cardiovascular, skin, and nerve regeneration. Finally, the review highlights the theranostic potential of exosomes followed by challenges, summary, and outlook.

## 1. Introduction

Extracellular vesicles (EVs) are a cluster of heterogeneous lipid bilayer particles of irregular shape and size (100–1000 nm), usually co-occur as exosomes and microvesicles, released by all living cells under both normal and pathological conditions (e.g., hypoxia, oxidative stress, etc.) [1]. The relatively smaller endosomal nanovesicles, measuring 30–150 nm, are referred to as exosomes, [2] which are distinct from exfoliated 200 to 1000 nm microvesicles or ectosomes, 1.0–5.0 µm dying cell fragments or apoptotic bodies, [3,4] and 50 nm non-membranous nanoparticles or exomeres, as shown in Figure 1A.

Exosome biogenesis occurs through inward budding or invagination of the plasma membrane, forming intraluminal vesicles (ILVs), which load cellular cargoes via the trans-Golgi network, mature into multivesicular bodies (MVBs), migrate toward the plasma membrane, and eventually fuse with it to release the ILVs as exosomes into the extracellular space [5]. It has been shown that all biological fluids, such as blood, urine, CSF fluid, semen, breast milk, saliva, and amniotic fluid, contain exosomes [3]. Additionally, pharmacokinetic studies demonstrate that when exosomes are administered orally, they undergo a wide range of biodistribution throughout the body, even penetrating deeper tissues after repeated administration. They undergo limited distribution in the liver, spleen, lungs, and gastrointestinal system and rapid clearance from the body after intravenous administration, compared to their administration through the intranasal route [5]. When inhaled through a nebulizer, they often perform better than synthetic lipid nanoparticles in terms of pulmonary bioavailability, distribution, and retention. They also reduce systemic side effects by localized delivery of drugs, mRNA, or proteins to the lung tissue [6]. Mass production of vascular endothelial growth factor A (VEGF-A)/bone morphogenetic protein 2 (BMP-2) mRNA-rich hASC-exosomes using the Track-Etched Membrane-Based Nanoelectroporation (TM-nanoEP) system and their delivery through PEGylated poly (glycerol sebacate) acrylate injectable hydrogel into rat femoral defects significantly increased their localization and bioavailability, which promoted bone regeneration via the angiogenic-osteogenic pathway [7]. Bioengineering approaches involving direct chemical modification or genetic engineering of exosome-secreting cells incorporating bioactive ligands can also improve localized delivery, specific cell or tissue targeting, and drug accumulation in target cells, thus improving efficacy and reducing off-target effects [8]. Exosome surface can be directly modified by conjugating CP05, RGE-peptide, BMSC-specific aptamer, RGD peptide, c(RGDyK) peptide, or genetic transfer (plasmid vectors) to exosome-secreting cells expressing various transmembrane proteins, either nonspecific (Lamp2b, tetraspanin) or receptor proteins (PDGFR, GPI, HER2, C1C2 domain of lactadherin) on exosome surface has significantly improved cell or tissue targeting. For instance, curcumin-loaded exosomes conjugated with c(RGDyK) peptide targeting ischemic brain injury [9], catalase delivery to neuronal cells in Parkinson’s disease [10], doxorubicin (DOX) delivery to tumor tissue via PH20 hyaluronidase-mediated degradation of hyaluronan matrix [11], or miR-140-loaded exosomes displaying high chondrocyte-affinity for osteoarthritis treatment [12] have produced excellent outcomes.

Although an optimized protocol for exosome isolation is unavailable, several studies mention procedures for isolating exosomes from different body fluids and conditioned media using techniques like ultracentrifugation [13], ultrafiltration [14], differential centrifugation (density gradient multi-step centrifugation) [15], immuno-capture using exosome-specific antibody-coated magnetic beads [16], polymer precipitation, and microfluidics [17], depending on the sample source (Figure 1B). This resulted in variable yields and purities of the isolated exosomes [15,16]. Many challenges are also encountered, such as ultracentrifugation producing low yield (~5%) of exosomes co-sedimented with nonspecific proteins, multi-step ultrafiltration involved risk of rupturing exosomal membrane, and differential centrifugation allowed co-precipitation of high-density lipoproteins (HDL) due to similar density with exosomes. In contrast, immuno-capture and microfluidic technologies involve high operating costs and lack the ability to mass-scale the production of clinical-grade exosomes. The isolated exosomes have been characterized by nanoparticle tracking analysis (NTA) and dynamic light scattering (DLS) for size distribution and concentration, surface zeta potential, sucrose gradient technique for calculating flotation density, electron microscopy for morphology, and microfluidic-based procedures for analyzing optical and electrical properties [3]. In addition, their surface biomarkers have been identified by a combination of liquid chromatography-mass spectrometry (LC-MS), matrix-assisted laser desorption/ionization (MALDI), high-resolution transmission electron microscopy (HR-TEM), reverse transcription polymerase chain reaction (RT-PCR), dodecyl sulfate-polyacrylamide gel electrophoresis (SDS-PAGE), western blot, etc., [2].

Exosomes are now well understood to be more than just pathological byproducts of tumorigenesis, inflammation, immunological responses, etc., or as alternative routes to lysosomal degradation for the disposal of cellular biowaste. They have many physiological roles, including cargo carriers, intracellular signaling, cell-cell communication, paracrine signaling, angiogenesis, extracellular matrix (ECM) synthesis, cell differentiation, etc., [18], which are usually mediated via intracellular cargo transport and surface ligands [19].

Mesenchymal stem cell (MSC)-derived exosomes display a variety of therapeutic effects, including cytoprotection, ischemic wound repair, and myocardial regeneration. In addition to mediating these effects, exosomal contents comprise phospholipids, specialized proteins, glycosphingolipids, and nucleic acids, which also act as biomarkers for secretory cells and aid in diagnosing several diseases [3]. Nearly 9769 surface and intraluminal proteins (>100 listed as biomarkers), 1116 lipids, 3408 mRNAs, and 2838 miRNAs were identified in exosomes using databases like Exocarta, Vesiclepedia, and EVpedia. These include proteins of cytosolic or membranal origin like endosomal sorting complex required for transport (ESCRT) proteins, ESCRT accessory proteins (TSG101, Alix), Rab, Ras, actin, tetraspanins, heat shock proteins, integrin, etc.; lipids like sphingolipids, glycerophospholipids, fatty acids, and cholesterol; and genetic materials like miRNA or miRs, siRNAs, rRNA, mRNA, tRNAs, Y RNAs, snRNAs, short hairpin RNAs, etc., [3,18,20]. Besides serving as natural nanocarriers, exosomes can also carry other therapeutic molecules (e.g., DOX, curcumin, and oxaliplatin) and deliver them to their targets. Labeling them with quantum dots (QDs), superparamagnetic iron oxide nanoparticles (SPION), or other fluorescent dyes (DiR, DiD, PKH67) allows their in vitro and in vivo monitoring.

Earlier reviews have mainly emphasized the application of exosomes as direct therapeutic agents or therapeutic nanocarriers. These reviews covered general topics like the nomenclature, biogenesis, composition, and biological activities of exosomes; the potential applications of exosomes as cell-free MSC-based therapy [21], cancer treatment [22], and their clinical utility for treating heart, kidney, liver, and lung diseases [23,24]; and issues like storage and large-scale production [25]. Others have compiled different engineering strategies for optimized delivery [26], in vivo imaging and tracking [27], and the therapeutic effects of exosomes, with a particular focus on neurological diseases [28], inflammatory and degenerative diseases, etc. [29]. Only a few studies have emphasized the role of MSC-exosomes in theranostics and tissue regeneration [20], which are exclusively covered in this review. After a general introduction, the therapeutic, regenerative, and theranostic roles of MSC-exosomes are discussed in the respective sections.

## 2. Applications

### 2.1. Therapeutic Role

#### 2.1.1. Inherent Therapeutic Effects

**Cancer**: Human MSC (hMSC) naturally tends to localize to tumor sites and secrete exosomes that influence biological processes such as metastasis, angiogenesis, and epithelial-mesenchymal transition (EMT) [30]. Owing to their innate biocompatibility, exosomes derived from MSCs (MSC-exosomes) have been used as a novel therapeutic tool for cancer treatment [31,32].

MSC-exosomes containing miRNAs, long non-coding RNAs (lncRNAs), proteins, etc., significantly inhibit cancer cell proliferation and promote apoptosis [33,34,35]. MSC-exosomes also reduce the potential for cell proliferation and metastasis in colorectal cancer models through the miR-100/mTOR/miR-143 pathway. Tumor suppressive activity was inversely correlated with miR-100 and mTOR expression [36] (Figure 2A). Bone marrow MSC-derived exosomes (BMSC-exosomes) effectively reduced pancreatic cancer cell viability and proliferation via intracellular miR-124 delivery [37].

**Cardiovascular diseases:** MSCs have garnered much attention because of their ease of isolation, high reproductive capacity, differentiation potential, and immunomodulatory qualities [41,42]. MSC therapy has emerged as a promising method for treating MI [43]. The cardioprotective effects of stem cell therapy mainly originate from paracrine effects rather than trans-differentiation [44,45]. MSC-exosomes contain various proteins and non-coding RNAs (miRNAs and lncRNAs) that act as paracrine mediators of cell-to-cell signaling. Ref. [46,47] By supplying pro-angiogenic factors to the damaged cardiomyocytes, MSC-exosomes promote angiogenesis and aid in healing cardiac wounds [48]. Angiogenesis was significantly promoted by elevated levels of miR-133a-3p, miR-221-3p, and miR-132 in MSC-exosomes (Figure 2B) [38,49,50]. Exosomes derived from miR-146a-modified adipose tissue-derived MSC (ASC) or miR-671-rich ASC-exosomes remarkably decreased MI by inhibiting apoptosis, inflammatory response, and fibrosis [51]. Similarly, the anti-apoptotic and cardioprotective effects of miR-126, miR-150-5p, miR-486-5p, and miR-125b in MSC-exosomes have been investigated in MI [52,53,54,55]. Preclinical studies revealed that MSC-exosomes containing miR-29b, miR-126, mir-133a, and miR-499 improved cardiac function by reducing cardiac fibrosis, infarct size, and myocardial cell death [56]. Intramyocardial injection of MSC-exosomes significantly improves the reperfusion rate after myocardial ischemia [57]. Nevertheless, clinical outcomes of exosomes in cardiac patients are significantly lacking [58].

**Neurological disorders:** Alzheimer’s disease (AD) is the primary cause of dementia, which occurs due to the formation of β-amyloid (Aβ) and Tau proteins responsible for plaque and tangle formation within the brain cells, causing progressive loss of neurons affecting cognition, memory, and language [59]. MSC-exosomes showed great promise for improving the condition owing to their anti-inflammatory, anti-apoptotic, and antioxidant effects, encouraging neo-vascularization, neuronal regeneration, and restoration of the disrupted blood-brain barrier [60]. MSC-exosomes reduced Aβ protein concentration, encouraging neurogenesis and neuroprotection against oxidative stress and inflammation, facilitating nutrient transport across the blood-brain barrier [61]. Enzymes like neprilysin (NEP) and insulin-degrading enzyme (IDE) naturally break down Aβ in the brain; a lack of these enzymes causes a rise in endogenous Aβ levels. Intravenous administration of MSC-exosomes loaded with NEP and IDE remarkably decreased Aβ plaque deposition in AD transgenic mice [62]. miR-133b-rich MSC-exosomes transferred to astrocytes and neurons promoted recovery of neural function [63]. hMSC-exosomes can significantly protect hippocampus neurons by reducing oxidative stress and minimizing Aβ mediated synaptic damage [64]. MSC-exosome-based therapies could successfully overcome challenges like apoptosis, necrosis, abnormal differentiation, and tumorigenesis caused by stress reactions and immune rejection following cell transplantation [65]. Neuroprotective molecule-loaded induced pluripotent stem cells (iPSC) derived exosomes (iPSC-exosomes) can ameliorate AD symptoms [66].

The effects of MSC-exosomes on attenuating neuropathology, improving cognitive performance, and delaying AD pathogenesis have been summarized by Wang et al. [60]. MSC-exosomes effectively reduced Aβ expression during an in vitro cell culture study while improving brain glucose metabolism, neurogenesis, and cognitive function in AD mice [67]. A phase I/II clinical trial using human ASC-derived exosomes (hASC-exosomes) showed significantly improved cognitive function in patients with AD over 36 weeks [68].

Parkinson’s disease (PD) is genetically and neuropathologically associated with protofibrils of the presynaptic neuronal protein α-synuclein, which induces disruption of cellular homeostasis and neuronal death. Several studies have established the beneficial effects of MSC-exosomes for treating PD, owing to their anti-inflammatory and neuroprotective effects, in addition to the modulation of α-synuclein aggregates. Nanosized (30–150 nm) exosomes effectively traverse the blood-brain barrier (BBB), providing better efficacy in PD treatment than conventional therapeutics [69,70]. hucMSC-exosomes demonstrated effective BBB penetration and improved behavioral function, neuronal damage, and dopamine concentration in the striatum after injection into a 6-hydroxydopamine-induced rat PD model. They also enhanced neuronal damage repair by lowering the release of inflammatory cytokines and autophagy [71,72]. Neuronal damage is also induced by microglial activation, and pro-inflammatory cytokines are inhibited by MSC-exosomes. Human umbilical cord MSC-derived exosomes (hucMSC-exosomes) dramatically reduce inflammation in addition to offering direct protection to neurons [73].

**Renal disease:** Since MSC-exosomes have healing effects similar to those of MSCs, they have been widely used in chronic kidney injury (CKI), particularly renal fibrosis and diabetic kidney disease, and have thus overcome the limitations of MSCs. However, the therapeutic effects of MSC-exosomes may vary depending on their origin.

hWJMSC-exosomes could effectively decrease renal injury and enhance renal function in a rat ischemia/reperfusion-induced acute kidney injury (AKI) model [74]. MSC-exosomes also offered protective effects against renal injury via inflammation regulation. In a study by Shen et al., MSC-exosomes showed higher expression of C-C motif chemokine receptor-2 (CCR2), which lowered the amount of its ligand (CCL2) and blocked its actions to increase macrophage infiltration in a mouse ischemia/reperfusion-induced AKI model [75]. Adipose MSC-derived exosomes can efficiently protect against ischemia/reperfusion-induced AKI [76]. By elevating the expression of miR-199a-3p in renal cells, BMSC-exosomes can inhibit the progression of ischemia/reperfusion-induced AKI [77]. MSC-exosomal cargo, including growth factors (GFs) and miRNAs, significantly affects renal function by enhancing autophagy and mitigating kidney fibrosis. BMSC-exosomes introduced into the kidney’s subcapsular area in streptozotocin (STZ)-induced diabetic rats reduced renal damage and inflammatory cell infiltration [78]. hucMSC-exosomes dramatically downregulated pro-inflammatory cytokines by transferring several GFs to renal TEC cell lines and human renal glomerular endothelial cell lines [79]. hucMSC-exosomes can also reduce renal fibrosis and restore renal function [80]. The anti-fibrotic effects were mainly derived from the Hippo and YAP signaling, TGF-β-Smad signaling, and podocyte mesenchymal-epithelial trans-differentiation pathways. Conversely, the anti-fibrotic properties of ASC-exosomes mainly result from anti-inflammatory activity via Sox9 overexpression and blocking TGF-β1-induced TEC transformation into a pro-fibrotic phenotype, apart from downregulating pro-fibrotic genes (collagen and TGF-β) [81,82]. Few studies have been published on using MSC-exosomes directly in human patients or lupus nephritis (LN) animal models. BMSC-exosomes increased mice’s survival rate by inhibiting IL-21 production and increased helper T-cell development, thereby decreasing LN [83]. Exosomal miRNA-20a reduced inflammation by promoting autophagy in a murine LN model [84].

**Hepatic disorder:** Treatment of benign and malignant liver diseases is practically challenging due to the limited liver regeneration capacity. Limited success has been achieved with partial hepatectomy, and sporadic instances of postoperative liver failure have been reported. MSC-exosomes successfully alleviated inflammation in various liver diseases by decreasing inflammatory regulators, such as interleukins (IL)-6 and IL-1β, as well as CD154, by CD4+ T cells [85]. In contrast, human endometrial MSC-derived exosomes (hEMSC-exosomes) restored liver function by suppressing hepatic cell apoptosis and inflammation at the injury site in a mouse acute liver injury model [86]. Systemic administration of MSC-exosomes attenuated liver damage in a lethal acute liver failure mouse model by accumulating at the injury site [87]. ASC- and hucMSC-derived exosomes have demonstrated potential in mitigating inflammasome-induced inflammation associated with acute liver failure by suppressing NLRP3 activation and downregulating TXNIP in hepatic macrophages [88,89].

**Pancreatic disorder:** Patients with type I diabetes mellitus (T1DM), an autoimmune pancreatic disease caused by pancreatic β-cell dysfunction and death, require lifelong therapy [90]. MSC-exosomes have demonstrated comparable therapeutic efficacy to MSCs in treating diabetes while avoiding the problems associated with MSCs [91]. ASC-exosomes could successfully ameliorate IFN-γ- and IL-17-mediated inflammation by increasing regulatory T cells in the spleen in the T1DM mice model [92]. Menstrual blood-derived MSC-exosomes (MMSC-exosomes) may also improve insulin synthesis by increasing the number of islets and insulin-producing β-cells with impaired function in a diabetic rat model [93]. Insulin resistance owing to β-cells dysfunction is the leading cause of type II diabetes mellitus (T2DM). In the STZ-induced T2DM rat model, hucMSC-exosomes not only improved glucose metabolism by reversing insulin resistance but also prevented β-cell death, restoring insulin secretion [94]. Additionally, BMSC-exosomes containing miR-29b-3p can effectively control age-related insulin resistance [95]. hucMSC-exosomes reduced pancreatic damage by accelerating repair, preventing acinar cell death, and managing systemic inflammatory responses in a rat model of traumatic pancreatitis, although the underlying mechanism is unknown [96].

#### 2.1.2. Therapeutic Nanocarriers

Exosomes have been identified as potential nanocarriers for complex therapeutics, such as anticancer drugs. The inherent nature of exosomes confers them the ability to enable intercellular communication [97]. MSCs are a vital source of exosomes and have been widely studied for their drug-carrying properties. Different therapeutic molecules can be delivered through exosomes via passive or active loading and other viable approaches, including transfection, in situ assembly, and synthesis [98]. Although incubation or passive loading is the simplest way to achieve exosomal drug loading via drug diffusion into donor cells, this technique is limited by variable drug loading and low loading efficiency. While transfection uses transfection reagents to facilitate the transfer of proteins, peptides, and nucleic acids through exosomes, it also suffers from similar low drug-loading issues. Sonication, electroporation, extrusion, freeze-thaw cycles, surfactant treatment, and dialysis are examples of active drug-loading techniques that deliver therapeutic molecules by rupturing the exosomal membrane and creating micropores; however, they may also cause exosome aggregation, protein denaturation, and exposure of immune cells. Drug loading into the exosome core or on their surfaces can be accomplished via in situ assembly and synthesis, which has limitations associated with preparation.

**Proteins and peptides:** Exosomes are recognized as potential delivery vehicles for delivering protein- and peptide-based therapeutics, considering their susceptibility to degradation. Exosomes have been effectively used to carry therapeutic proteins, such as vesicular stomatitis virus glycoprotein (VSVG), [99] vaccination against SARS coronavirus, [100] or ornamental membrane proteins, such as TNF-related apoptosis-inducing ligand (TRAIL), which target cancer cells [101]. Although exosomes present promising delivery methods for protein and peptide therapies, issues like efficient drug loading, immunogenicity, dose selection, large-scale manufacturing, and purity must be resolved [102].

**Nucleic acids:** Exosomes can be excellent DNA and RNA nanocarriers for gene therapy since nucleic acids are highly prone to degradation [103]. Using a membrane engineering technique, exosomal membranes and liposomes were fused to create a hybrid gene delivery platform that could distribute hydrophilic cargo inside exosomes and exogenous hydrophobic lipids to target cells [104]. NanoMEDIC, an exosomal ribonucleoprotein delivery system, was developed for in vivo genome editing by packaging single guide RNA (sgRNA) and CRISPR-single bond Cas9 protein through HIV-derived RNA packaging signal and HIV-derived Gag protein, respectively [105].

**Drug molecules:** BMSC-exosomes loaded with gelactin-9 siRNA by electroporation or surface modified with oxaliplatin significantly improved tumor-targeting efficacy without significant organ toxicity [106]. Intravenously administered paclitaxel-loaded hucMSC-exosomes effectively treated breast cancer in mice, showing marked inhibition of primary tumor growth and distant organ metastases [107]. DOX-loaded MSC-exosomes via the electroporation technique led to targeted drug delivery to HER2+ breast cancer cells; thus, it can potentially minimize side effects on normal cells while enhancing therapeutic efficacy. Ref. [108] Compared to free drugs, DOX-loaded BMSC-exosomes significantly improved cellular uptake and antitumor effects in the MG63 osteosarcoma cell line; hence, they are helpful for osteosarcoma therapy. Ref. [109] Similarly, MSC-exosome-based cabazitaxe delivery could be an effective strategy for treating drug-resistant oral squamous cell carcinoma. Ref. [110] BMSC-exosomes mediated the delivery of the anticancer drug norcantharidin, improved cellular uptake, and reduced tumor cell proliferation compared to free drug alone [111]. As shown in Figure 2C, Pretreated Wharton’s jelly derived MSC (WJMSC)-exosomes remarkably improved the cellular activity of dopaminergic neurons, mitochondrial function, and cell survival by blocking 6-hydroxydopamine-induced apoptosis, representing a promising strategy for PD therapy via reduced α-synuclein aggregation [39].

Natural plant- and animal-derived exosomes are often modified (internally or externally) to introduce various functional groups on their surface or subjected to charge alteration, thereby promoting the absorption of therapeutic molecules through electrostatic interactions or covalent conjugation [112]. The “postinsertion” approach decorates the exosome surface with selective molecules via PEG, enhancing cell selectivity and circulation time [113]. Recently, a nanofilm comprising supramolecular complexes of ferric ions (Fe^3+^) and tannic acid was created on the exosome surface to facilitate the adsorption of DOX and conjugation with folic acid, enabling specific localization of the drug-loaded exosomes into cancer cells (Figure 2D) [40]. Moreover, conjugation of the central nervous system-specific rabies viral glycoprotein (RVG) peptide on the surface of MSC-exosomes enabled greater cortex and hippocampus targeting [114]. Table 1 outlines some completed and ongoing clinical trials employing therapeutic exosomes produced by MSCs of distinct origins and administered through different routes to treat various diseases. However, further studies are required to ensure risk-free therapy in clinical settings.

### 2.2. Regenerative Role

#### 2.2.1. Bone and Cartilage Regeneration

Congenital defects, trauma, and aging cause damage to load-bearing tissues like bone and cartilage. Bone regeneration is a multi-step (immunomodulation, osteogenesis, and remodeling) and multifactorial process in which bone tissue engineering can be effectively addressed using scaffolds/hydrogels, cells, and growth factors. Despite significant advancements in bone tissue engineering, unfavorable safety and efficacy concerns remain. Joint cartilage cannot self-regenerate following injury or joint conditions, such as rheumatoid arthritis and osteoarthritis, where tissue engineering can play an immense role. Due to the constraints of MSC harvest, growth, and storage, the paradigm for bone and cartilage tissue regeneration is changing from “whole cell therapy” to “cell-fee exosomes”, reducing ischemic tissue damage by promoting endogenous cell repair and angiogenesis [115].

Exosomes play a crucial role in bone remodeling, allowing osteoblasts, osteocytes, and osteoclasts to communicate with and maintain the bone tissue [116]. BMSC-exosomes could potentially increase osteoblastic differentiation by upregulating osteogenic-related genes (miR-196a, miR-27a, and miR-206) and facilitating the recovery of in vivo calvarial defects [117]. Modified MSC-exosomes enriched with miR-135b and TGF-β1 successfully inhibited osteoclastogenesis in an osteoarthritic mouse model without affecting the microarchitecture of the subchondral bone [118]. Exosomes are commonly included in scaffolds used for bone tissue regeneration, because they contain several bioactive components. As shown in Figure 3A, exosomes secreted by human-induced pluripotent stem cell (hiPS)-derived MSC (hiPS-MSC-exosomes) and supplied via β-tricalcium phosphate (β-TCP) significantly enhanced osteogenic differentiation of hBMSC via the PI3K/AKT signaling pathway [119]. hEMSC-exosomes delivered through a porous bioglass scaffold remarkably improved osteogenesis in rat critical-size bone defects [120]. Similarly, bone regeneration was greatly enhanced by hASC-exosomes immobilized on polydopamine-coated PLGA scaffolds [121]. As vascularization is crucial for bone regeneration, decalcified bone matrix (DBM) functionalized with MSC-exosomes significantly enhances bone regeneration through dual osteogenic and pro-angiogenic activities [122]. miRNA expression levels in exosomes can be analyzed to assess bone health or disease progression [123]. The levels of Wnt-related proteins or other osteogenic markers in exosomes can provide an impression of osteogenic potential in patients with bone disorders [124]. MSC-exosomes have been shown to effectively suppress pro-inflammatory cytokines, such as IL-1β and TNF-α, and increase the anti-inflammatory factor TGF-β [125]. By measuring the levels of these cytokines in patient samples, clinicians can evaluate the inflammatory status during the healing process [126]. miR-199b, Let-7, and miR-135b levels in MSC-exosomes can be assessed to determine the state of osteogenic differentiation. Elevated levels of Let-7 and miR-199b could be indicative of osteogenesis, while upregulated miR-135b may indicate impaired differentiation [123,127,128]. MSC-exosomes can also regulate angiogenesis by an unknown mechanism [122]. miR-21-rich MSC-exosomes possessing anti-apoptotic and osteoblastogenic properties greatly improved rat femoral defect healing [129,130]. Therefore, monitoring miR-21 expression may provide insights into both healing and therapeutic interventions.

Since both hard and soft tissues are damaged in maxillofacial injuries, bone grafts provide a framework for osteoconduction in craniofacial defects, promoting bone healing through osteoblastic differentiation [132]. MSC-exosomes can enhance fracture healing in CD9 mice, especially in those with decreased exosome production and delayed callus development [125]. Studies have shown that exosomal miRNAs regulate osteoblastic differentiation via the Wnt and PI3K/Akt pathways [133]. Exosomes derived from osteoblastic cells, [134] dendritic cells, [135] and monocytes [136] could also promote osteoblastic differentiation of MSC. miRNAs derived from MSC-exosomes are frequently linked to the regulation of osteoblast activity and bone formation [137]. Exosomes secreted by hASC after osteogenic induction demonstrated greater osteoinductive effects and enhanced bone healing in mouse calvaria-defects [121]. Engineered stem cell-derived exosomes demonstrated superior effects on oral and maxillofacial wound treatment than conventional therapy. iPS-MSC-exosomes delivered through a β-TCP scaffold stimulated osteoblast differentiation in vitro, in addition to promoting regeneration of rat critical-sized bone defects [138].

Intra-articular injection of MSC-exosomes exerted chondroprotective effects in a collagenase-induced OA model with higher expression of chondrogenic markers and inhibition of catabolic and inflammatory markers, causing chondrocyte apoptosis and monocyte activation [139]. Further studies have revealed that hiPS-MSC-exosomes can significantly promote chondrocyte migration and proliferation, deposition of cartilage matrix, and reduce cell death [140]. They can also decrease inflammatory gene expression in temporomandibular joint-OA, passivating the early inflammatory response [141,142]. hiPS-hESC-exosomes delivered through an acellular tissue patch made of a photo-induced imine crosslinking (PIC) hydrogel led to good integration with the native cartilage matrix while increasing the chondrocyte population at the injury site [143]. On the other hand, BMSC-exosomes delivered through a 3D printed scaffold made of cartilage matrix and gelatin methacrylate fully restored injured cartilage by improving chondrocyte migration [144]. hucMSC-exosomes inhibited apoptosis, promoting proliferation, migration, and matrix formation by chondrocytes by significantly increasing the expression of chondrogenesis-related mRNA and protein [145]. Therefore, MSC-exosomes can be used as a cell-free strategy for the regeneration of cartilage defects as an alternative to cell therapies.

Osteoarthritis is a joint degenerative disorder resulting from trauma, autoimmune responses, or hereditary factors leading to pain, inflammation, and loss of function. Osteoarthritis, ECM degradation, and chondrocyte hypertrophy generate an inflammatory state [146]. According to Cosenza et al., MSC-exosomes injected into the articular cavity of mice could prevent osteoarthritis [147]. Intra-articular injection of hEMSC-exosomes demonstrated complete cartilage and subchondral bone regeneration in critical-sized osteochondral defects in rats [148]. Acellular cartilage ECM-derived scaffolds incorporated with hucMSC-exosomes considerably improved osteochondral regeneration after subcutaneous implantation in a rat osteochondral defect model [149]. Similarly, hWJMSC-exosomes incorporated into acellular cartilage matrix scaffolds encouraged osteochondral regeneration by reducing inflammation in the joint cavity and promoting the deposition of cartilage ECM [150]. hESC-exosomes potentially reduce cartilage damage and matrix breakdown in an osteoarthritic mouse model [151]. hESC-exosomes exhibit great potential to effectively repair osteochondral defects by stimulating cell migration and proliferation, and regulating apoptosis and immune responses [152]. Therefore, MSC-exosomes can be used as a cell-free strategy for osteochondral defects as an alternative to cell therapies.

#### 2.2.2. Angiogenesis

Angiogenesis involves vascular cell activation, migration, and proliferation, and the emergence of new vessels from old vasculature via interactions between cells, soluble factors, and ECM [153,154]. Vascularization is crucial for gas and nutrient exchange during bone formation. MSC-exosomes may influence vascular development mediated by miRNAs or lncRNAs.

BMSC-exosomes greatly supported angiogenesis by stimulating the migration, proliferation, and tube formation of ECs [155]. The inclusion of rat BMSC-exosomes into tissue-engineered DBM scaffolds induced simultaneous pro-osteogenic and pro-angiogenic activities, which eventually aided bone regeneration [122]. hucMSC-exosomes integrated with 3D-printed 3D-printed silk fibroin/collagen I/nano-hydroxyapatite (SF/COL-I/nHA) composite scaffolds significantly promoted alveolar bone defect repair through the osteogenic-angiogenic pathway [156]. Pluronic F127 hydrogel containing hucMSC-exosomes caused human umbilical vein endothelial cells (HUVEC) to express more CD31, Ki67, VEGF, and TGFβ-1 and promoted angiogenic activity [157]. Hypoxia-inducible factor 1-alpha-modified rat BMSC-exosomes increased pro-angiogenic factors that encouraged HUVEC migration and proliferation, fostering neovascularization [158]. MSC-exosomes overexpressing pro-angiogenic miRNA-21-5p in response to inflammatory cytokines (TNF α and IL-6) and vascular cell adhesion molecule-1 significantly promoted in vitro cell proliferation and pro-angiogenesis in the Chick Chorioallantoic Membrane (CAM) assay, which in turn facilitated ischemic tissue repair in a diabetic rat model [159]. rBMSC-exosomes delivered through a polyethylene glycol maleate citrate (PEGMC)/β-TCP hydrogel promoted both osteogenic and angiogenic activities [160]. Exosomes enriched in miR-21-5p from preconditioned ASC enhanced pro-angiogenic effects in HUVECs, thereby neo-bone formation in a mouse osteoporotic defect model [161]. Hypoxia-preconditioned exosomes released by human exfoliated deciduous teeth (SHED)-derived MSC significantly promoted angiogenesis in rat calvarial defects [162]. Similar angiogenic effects were found with exosomal HMGB1 released by hypoxic-preconditioned BMSC [163]. Another study revealed that ASC-exosomes promote the migration and activity of vascular ECs by dramatically upregulating angiogenesis-associated proteins (VEGF, FILK1, and ANG1) and downregulating angiogenesis inhibitors (TSP1 and VASH1) [164]. Similarly, BMSC-exosomes accelerated vascularization and regeneration in rat diabetic wounds by upregulating miRNA-211-3p and activating AKT/eNOS signaling [165].

#### 2.2.3. Periodontal Tissue Regeneration

Periodontitis is characterized by a chronic inflammatory state caused by gradual deterioration of periodontal tissues, including cementum, periodontal ligament (PDL), alveolar bone, and gingival tissue, due to microbial plaque development. While therapeutic exosomes derived from MSC significantly impact periodontal repair and regeneration, exosomes carrying proteins and nucleic acids may serve as biomarkers for periodontal disease [166]. Exosomes produced by BMSCs [167], hASCs [168], human placenta-derived MSCs [169], and periodontal ligament stem cells [170] encouraged in vitro migration, lumen development, and proliferation of endothelial cells. Moreover, hiPS-MSC and hASC-derived exosomes delivered through β-TCP and polydopamine-coated PLGA scaffolds led to faster generation in osteoporotic rats and mice calvarial defects [121,138,171]. The transfer of exosomal miRNAs, such as miR-214-3p, miR-183-5p, miR-196a, and growth factors (e.g., BMP and TGFβ1) significantly enhances bone formation [172].

#### 2.2.4. Skin Regeneration

The skin, the body’s largest organ, comprises three distinct layers- the epidermis, dermis, and hypodermis- contributing to protective and sensory activities. GFs, cytokines, and chemokines are key players that regulate different stages of wound healing (inflammation, cell migration, granulation, angiogenesis, and neotissue formation) [173,174]. MSC-exosomes possessing regenerative and immunomodulatory properties can significantly promote wound healing [175,176]. MSC-exosomes can alleviate inflammatory states by substantial upregulation of miR-132, promoting M2 macrophage polarization and reduced cytokine production [177,178]. Elevated levels of miR-132, miR-21, and miR-126 in MSC-exosomes, along with other reported miRs, such as miR-23a, miR-29a, and miR-29b, can promote wound healing, mainly due to their roles in macrophage activation, angiogenesis, and TGF-β signaling [179]. Several studies have focused on exosome therapy to promote skin regeneration. ASC-exosomes loaded in an alginate-based hydrogel significantly improved wound healing in a rat model [180]. An injectable self-healing hydrogel composed of Pluronic F127, oxidative hyaluronic acid (OHA), and poly-ε-L-lysine releasing ASC-exosomes shortened the healing time by increasing cell proliferation, tissue generation, and re-epithelialization in a mouse diabetic wound model [181]. Chitosan/silk scaffolds loaded with gingival MSC-exosomes enhanced re-epithelialization and ECM deposition/remodeling in the defect region, promoting skin regeneration in a diabetic rat skin model [182].

#### 2.2.5. Nerve Regeneration

The peripheral nervous system innervates almost all tissues and organs for motor and sensory functions. Peripheral nerve injury (PNI) is a complicated neurodegenerative disease that affects blood vessels, Schwann cells (create the myelin sheath), and nerve fibers, resulting in sensory dyskinesia or autonomic limb dysfunction. MSC-exosomes, as carriers of paracrine factors, efficiently control the microenvironment and neuronal cell activity [78]. A 3D composite nerve conduit loaded with hucMSC-exosomes produced a native tissue-like microenvironment that encouraged axonal growth and nerve regeneration in a rat peripheral nerve injury model [183]. In a rat sciatic nerve injury model, nerve conduits with electrospun PLGA in their outer shell on a collagen-hyaluronic acid sponge core demonstrated enhanced axonal growth and motor function, nerve regeneration, and remyelination. These effects were further improved by the incorporation of hucMSC-exosomes into the conduits [184]. The stiffness of hydrogels laden with exosomes significantly impacted the healing of PNI. In a rat sciatic nerve injury (SNCI) model, relatively softer hydrogels promoted the quick release of exosomes, improving nerve regeneration, inhibiting macrophage infiltration, and releasing pro-inflammatory cytokines [185]. Alginate scaffolds containing hucMSC-exosomes reduced pro-inflammatory cytokines and increased anti-inflammatory and neuroprotective factors (GDNF, IL-10, and myelin basic protein), reducing pain in rats due to nerve injury [186]. hMSC-exosomes immobilized in a peptide-modified adhesive hydrogel dramatically reduced nerve injury in a rat long-span spinal cord transection model by lowering oxidative stress and inflammation [131] (Figure 3B). Due to the exosome’s antioxidative, anti-inflammatory, and neurotrophic qualities, the hucMSC-exosome immobilized scaffold significantly recovered nerve injury in the SD rat spinal nerve ligation pain model [187].

Peripheral nerve regeneration can be facilitated by exosomes by increasing the migration, proliferation, and myelination of Schwann cells [188,189]. The quantity and width of nerve fibers were greatly increased by the BMSC-exosome-functionalized PDA-modified chitosan conduit, in addition to myelination [190]. An ideal environment for peripheral nerve regeneration and functional recovery can be created by establishing a vascular network [191]. In rats with acute spinal cord injury, hucMSC-exosomes encapsulated in fibrin gel efficiently reduced inflammation and oxidative stress while promoting nerve tissue regeneration [192]. Spinal cord injury (SCI) is difficult to treat due to the lack of self-regenerative ability. In an early SCI model, MSC-exosomes integrated into the electroconductive GelMA/polypyrrole (PPy) hydrogel network, which released exosomes for up to 14 days, were able to successfully bridge the transected spinal cord and partially restore endogenous electrical signal transmission [193]. External trauma causes structural damage and/or functional impairment of the brain tissue in traumatic brain injury (TBI). BMSC-exosomes added to HA-collagen hydrogel mimicked neurogenesis, angiogenesis, and functional recovery of neurons in TBI [194]. 3D printed collagen/chitosan scaffolds loaded with brain-derived neurotrophic factor (BDNF)-pretreated hucMSC-derived exosomes significantly enhanced the reparative effects by remodeling neural networks and remyelinating the implanted site [195]. Similarly, 3D-printed collagen/silk fibroin scaffolds containing hypoxia-induced hucMSC-exosomes significantly promoted neural regeneration and angiogenesis in a beagle TBI model [196].

#### 2.2.6. Liver Regeneration

MSC-exosomes can significantly improve the self-healing capacity of the liver. Hydrogels loaded with MSC-exosomes may facilitate the rapid distribution of exosomes and accelerate the repair of injured liver tissue [197,198]. However, chronic liver injury often leads to inflammation and fibrous scarring. Hepatocyte apoptosis and fibrosis were significantly reduced by systemic delivery of injectable hydrogel through the peritoneal cavity, prolonging the release of pre-loaded MSC-exosomes, as well as bioavailability at the target site [199].

### 2.3. Theranostic Role

The most popular use of fluorescence imaging is to monitor the distribution of exosomes, which are typically labeled with fluorescent probes. These probes frequently encounter issues such as photobleaching and shallow tissue penetration. Exosomes can be employed as nanocarriers to help the penetration of drugs and image contrast agents through the blood−brain barrier (BBB), thereby monitoring brain diseases (stroke, AD, and PD). As shown in Figure 4A, BMSC-exosomes labeled with gold nanoparticles (GNP) showed preferential uptake and accumulation in neural cells and brain tissue, as revealed through CT imaging [200]. Targeted MRI imaging and treatment of brain gliomas were demonstrated by curcumin-loaded exosomes linked with SPIONs via neuropilin-1-targeted peptide (RGE) peptide (Figure 4B) [201]. Additionally, radionuclide-labeled exosomes allow for 3D PET or SPECT imaging of different organs and tissues through selective localization [28]. Glucose-coated fluorescent semiconductor polymer dots in the second near-infrared window (NIR-II), which are used to label and track MSC-exosomes, not only allow for quick recovery of postoperative liver function but also inhibit inflammatory responses, apoptosis, and cell proliferation, making them promising for liver tissue regeneration [202]. A versatile theranostic platform for colorectal cancer was made possible by electroporatic loading of DOX into MSC-exosomes and functionalization with a carboxylic acid-end MUC1 aptamer (DOX@exosome-apt), which allowed for fluorescence imaging via selective delivery to MUC1-positive cancer cells and inhibition of tumor growth in BALB/c mice [203]. BMSC-exosomes loaded with curcumin and functionalized with c(RGDyK) peptide allowed for targeted delivery to the ischemic brain of middle cerebral artery occlusion (MCAO) mice, lowering the inflammatory response and cellular apoptosis; therefore, they have the potential to be used as a theranostic for the treatment of ischemic stroke [9]. Labeling MSC-exosomes with fluorescent dyes (DiD, DiR, and PKH26) can allow optical imaging using an in vivo imaging system (IVIS) and confocal microscopy, whereas labeling with gadolinium (Gd^3+^) lipid and ferritin heavy chain (FTH1)-lactadherin (LA) can build MRI contrast [204,205,206,207,208].

Exosomal RNA profiles can indicate the presence of liver malignancies. Elevated levels of exosomal miR-21 have been investigated for their potential as a non-invasive diagnostic biomarker for acute liver diseases and hepatocellular carcinoma [209,210]. Hepatocyte-exosome-mediated transfer of neutral ceramidase and sphingosine kinase-2 at the regeneration site may serve as a valuable diagnostic tool for assessing liver health [211]. In this context, MSC-exosomes may be a helpful diagnostic and treatment tool for liver damage, as demonstrated in CCl_4_-induced liver injury [212].

Administration of BMSC-exosomes can enhance cardiac repair in myocardial infarction (MI) patients by downregulating CD68 expression [213]. It has been shown that the intake of MSC-exosomes decreases CD68 expression, which is associated with a reduction in inflammation and macrophage activation. Exosomes obtained from miR-146a-modified ASC suppressed inflammation and apoptosis; therefore, miR-146a levels could serve as a potential biomarker for the severity of acute myocardial infarction (AMI) and the effectiveness of ASC-exosome therapy [51]. Exosomes secreted by atorvastatin-pretreated MSCs were found to be rich in lncRNA H19, which effectively reduced cardiac dysfunction by decreasing IL-6 and TNF-α levels; thus, lncRNA H19 levels could be used as a diagnostic marker for angiogenesis and apoptosis associated with cardiac dysfunction and infarction [214]. miR-93-5p-rich exosomes derived from ASC could effectively treat MI, indicating their diagnostic and therapeutic potential in MI [215]. Levels of miR-let7, miR-182, miR-146a, miR-181b, and miR-126 in MSC-exosomes can be negative indicators of inflammatory status, fibroblast proliferation, and macrophage polarization in cardiac diseases; hence, they are useful as potential biomarkers for diagnosing atherosclerosis [216,217,218]. MSC-exosomes are also effective in ventricular dilation by reducing inflammatory cytokines and promoting M2 macrophage polarization, indicating their theranostic potential in dilated cardiomyopathy [219].

## 3. Challenges

Despite showing promise for essential uses like tissue regeneration, theranostics, and disease diagnostics, much work needs to be done before they can be used in clinical settings. The current obstacles to large-scale production, such as low yield, heterogenicity, targeted delivery, storage, and lack of standards to ensure quality, safety, efficacy, stability, and reproducibility, should be addressed.

Exosomes’ therapeutic effects and tissue regeneration in the complex wound milieu are always unpredictable due to the lack of knowledge regarding their interactions with specific cells or tissues, cellular uptake mechanisms (endocytosis or direct fusion), and short half-life due to rapid clearance [143,220]. The quantifiable metrics to identify the cellular origin and their suitable isolation and purification strategies can help maintain exosome quality [65]. Benchtop techniques like ultracentrifugation and density gradients have a limited yield, purity, and integrity, making it difficult to scale up exosome synthesis to produce enough for clinical usage without compromising quality and functionality [221]. Due to the lack of standardized protocols, isolating and characterizing the heterogeneous population of exosomes and microvesicles that come from various cellular origins and biogenesis pathways with unique functional properties is difficult. This significantly reduces their reproducibility in clinical and research settings. Pure exosomes can be obtained using antibody-based immunoaffinity techniques; however, these methods are expensive and produce low yields [222].

Depending on the origin, function, and niche of MSC and external factors, there is always a dynamic interplay between exosome production and content variability, which poses a severe obstacle to the clinical translation of MSC-exosomes. Exosomes typically release their payload into the cytoplasm through endosomal escape; however, lysosomal degradation of their contents eventually reduces their therapeutic effects. Environmental factors such as freeze-thaw cycles, osmotic pressure, pH, and hypoxic state, greatly influence MSC-derived exosomal contents’ stability, particularly the bioactive proteins and miRNA, which are also responsible for variable therapeutic outcomes [106,223]. Monitoring morphological traits, zeta potential, size distribution, concentration, surface biomarkers, etc., may be crucial for efficiently loading drugs into exosomes with batch-to-batch consistency and purity while preserving their stability [224].

Clinical studies and approvals have been significantly delayed, as exosome-based therapeutics must undergo extensive review to fulfill safety and efficacy standards, especially immunogenicity, which presents considerable regulatory obstacles. Since the US Food and Drug Administration (FDA) has not officially classified exosomes, their regulatory status in the biopharmaceutical sector is unknown [225]. Inadequate quality assurance procedures are in place to ensure exosome efficacy, safety, and reproducibility by establishing consistent release parameters, which require rigorous validation before using them in a clinical setting [226,227,228,229]. Although, the International Society for Extracellular Vesicles (ISEV) has published guidelines “Minimal Information for Studies of Extracellular Vesicles (MISEV)” (latest version: MISEV2023) on nomenclature, sample collection, separation, characterization, and functional studies for conducting a systematic EV research, even in clinical settings. The International Society for Cellular and Gene Therapies (ISCT) and the Society for Clinical Research and Translation of Extracellular Vesicles Singapore (SOCRATES) are dedicated to standardizing protocols for translating bench-to-bedside research involving EVs, cells, and genes without compromising patient safety and efficacy.

More importantly, MSC-exosomes may also carry the risk of tumor growth, immunosuppression, and drug resistance. For example, hASC-exosomes may increase the growth and invasion of breast cancer cells by activating Wnt signaling pathways, which accelerates tumor progression. Similarly, BMSC-exosomes containing miR-221 may promote the growth of gastric cancer cells. MSC-exosomes, particularly those carrying miRNA-21-5p, can create an immunosuppressive environment by polarizing M2 macrophages and suppressing T-cell activity. Such an environment leads to anti-inflammatory storms by increasing the expression of anti-inflammatory cytokines (TGF-1 and IL-10) and decreasing pro-inflammatory cytokines (IL-6, IL-1B, IL-12p40, and TNFα), which impedes the body’s immune response against tumors and permits the growth of cancer cells [230]. Drug resistance is another issue with MSC-exosomes; in particular, those containing miRNA-222/223 or miRNA-23b make breast cancer resistant to the proteasome inhibitor docetaxel, and BMSC-exosomes carrying upregulated mRNAs of PSMA3 and PSMA3-AS1 drug transporter proteins may cause resistance in multiple myeloma cells [230]. Therefore, further investigation is necessary to fully understand the potential role MSC-exosomes may play in risk-free clinical treatment.

## 4. Summary and Outlook

Exosomes are naturally occurring nanovesicles that carry various exosomal cargoes that are crucial for intracellular signaling, cell-cell communication, paracrine signaling, angiogenesis, ECM synthesis, cell differentiation, and other processes. They can either cause disease or promote healing. This present study reviewed the widespread application of MSC-exosomes as theranostic agents, therapeutic nanocarriers, and promoters of tissue regeneration.

The direct therapeutic effects of MSC-exosomes include tumor suppression, cytoprotection, ischemic wound repair, hepatoprotection, improvement of impaired renal function, bone, skin, nerve, periodontal tissue, and myocardial regeneration. They can also help deliver therapeutic agents (like cancer chemotherapeutics), occasionally with a diagnostic probe for concurrent monitoring and treatment. In this context, a strategic alteration of exosomes could be beneficial for immobilizing therapeutic or cell/tissue targeting molecules for enhanced localization, cargo enhancement, controlled biodistribution and release, and biological half-life, ultimately enhancing their therapeutic efficacy. Implementing good manufacturing practice (GMP) for exosome isolation, carefully selecting biomarkers for differentiating functional exosomes from non-functional ones, and optimizing dosage, administration route, and therapeutic regimen may contribute to higher therapeutic efficacy. Most exosome research has been cell-based; therefore, biosafety and long-term therapeutic efficacy should be monitored in vivo. Pilot-scale, multicentric, long-term clinical studies should be conducted to achieve clinical usability status; however, strict quality control of exosome products must be established, defining consistent release criteria based on size, surface markers, cargo, etc.

## Figures and Tables

**Figure 1 cells-13-01956-f001:**
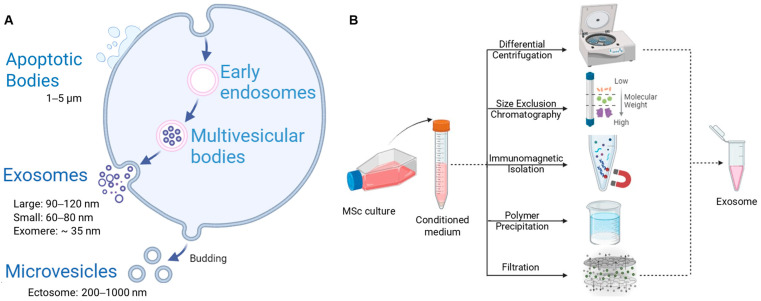
(**A**) Schematic illustrations of exosome biogenesis and (**B**) their different isolation techniques. (figure generated via biorender.com).

**Figure 2 cells-13-01956-f002:**
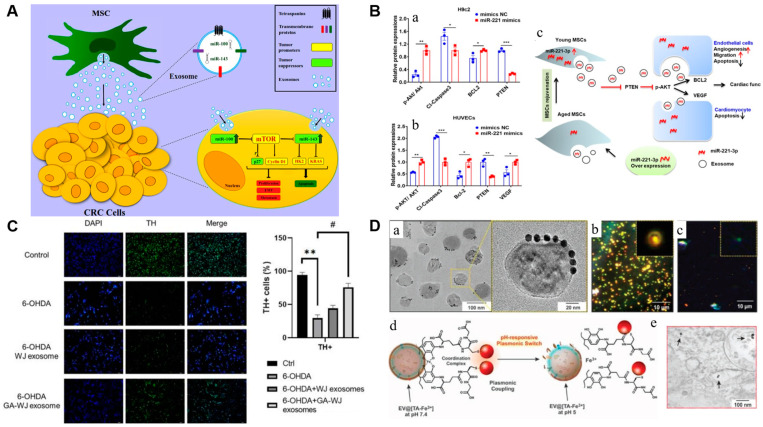
Therapeutic application of MSC-exosomes. (**A**) Schematic diagram representing the molecular mechanisms of MSC-exosomes involved in cell proliferation and metastasis in CRC cells. Ref. [36] Copyright© 2022, Elsevier, Amsterdam, Netherlands. (**B**) A quantitative study of the expression of Akt, p-Akt, Cleaved caspase-3, Bcl-2, and PTEN in (**a**) H9c2 cells and (**b**) HUVECs (*n* = 3) revealed that the PTEN/Akt signaling pathway was responsible for the recovery of cardiac injury mediated by exosomal miR-221-3p. * *p* < 0.05; ** *p* < 0.01; *** *p* < 0.001. (**c**) The suggested working model of this study. Ref. [38] Copyright© 2020, The Authors, Frontiers Media S.A, Lausanne, Switzerland. (**C**) Images after tyrosine hydroxylase (TH) staining and plot showing that exosomes derived from GA-pretreated WJMSCs were able to protect TH-positive cells against 6-hydroxydopamine (6-OHDA)-induced damage. ** *p* < 0.01 vs. control and # *p* < 0.05 vs. 6-OHDA groups. Ref. [39] Copyright© 2023, Impact Journals, LLC, NY, USA. (**D**) (**a**) TEM images of exosomes surface modified with GSH-AuNPs (**b**), (**c**) Dark-field microscopic images of single exosomes in PBS at pH 7.4 and 5, respectively. (**d**) Schematic representation of pH-responsive plasmonic switch of GSH-AuNPs-functionalized exosomes. (**e**) TEM images of fixed and sectioned cells revealing the cytoplasmic localization of GSH-AuNP-functionalized exosomes (black arrow). Ref. [40] Copyright© 2018, WILEY-VCH Verlag GmbH & Co. KGaA, Weinheim, Germany.

**Figure 3 cells-13-01956-f003:**
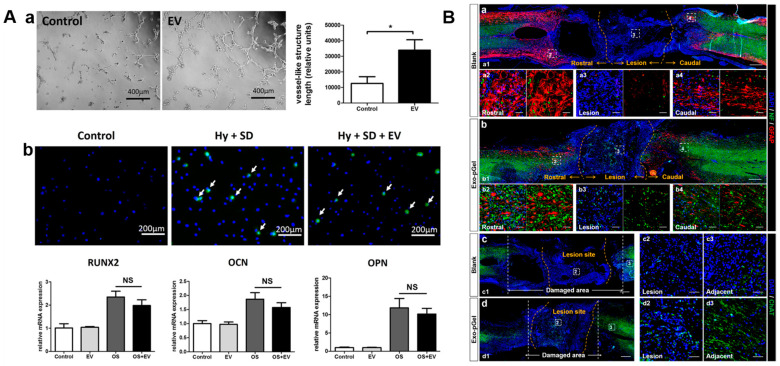
Regenerative potential of MSC-exosomes. (**A**) MSC-exosomes promote dual angiogenic-osteogenic activity by stimulating (**a**) proliferation, migration, and tube formation of HUVECs (*n* = 3). * *p* <  0.05; (**b**) proliferation, apoptosis, and osteogenesis of MSCs. White arrows and green dots indicate the TUNEL-positive cells. Plots represent qRT-PCR analysis data of osteogenic-related gene (Runx2, OPN, and OCN) expression (*n* = 3). NS, not significant at the level of 0.05 [122]. Copyright© 2017, The Author(s), Springer Nature, London, U.K. (**B**) Effect of hucMSC-exosomes on spinal cord regeneration. (**a**,**b**) Representative images showing neurofilaments (green) and glial fibrillary acidic protein (red), where yellow dashed lines indicate the transected area, and arrows indicate the rostral (**a2**,**b2**), defect (**a3**,**b3**), and caudal (**a4**,**b4**) regions. (**c**,**d**) After 28 days, fluorescent immunostaining reveals the precise density and distribution of choline acetyl transferase (ChAT, green) in the defect (**c2**,**d2**) and surrounding (**c3**,**d3**) tissues, with yellow and white dashed lines indicating the transected and damaged areas. Scale bar, 500 μm (**a1**–**d1**), 50 μm (**a2**–**a4**,**b2**–**b4**,**c2**,**c3**,**d2**,**d3**) [131]. Copyright© 2020, American Chemical Society, Washington, DC, USA.

**Figure 4 cells-13-01956-f004:**
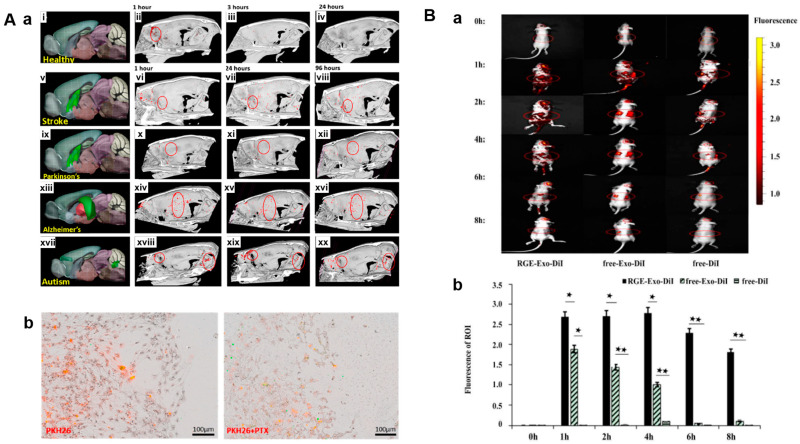
(**A**) Theranostic properties of gold nanoparticle (GNP)-labeled MSC-exosomes (**a**) accumulation of GNP-labeled MSC-exosomes in normal and diseased brains (Stroke, Parkinson’s disease, Alzheimer’s disease, Autism) examined by CT imaging (**b**) higher fluorescence imaging signal in PKH26-labeled MSC-exosomes (left) compared to paclitaxel-loaded MSC-exosomes (right) 96 h post-treatment, indicating drug-induced neuronal cell death [200]. Copyright© 2019, American Chemical Society, Washington, DC, USA. (**B**) Application of SPIONs and curcumin (Cur)-loaded exosomes for antitumor therapy and simultaneous imaging. (**a**) Better glioma targeting was attained by conjugating the exosomal membrane with neuropilin-1-targeted peptide (RGE), as revealed by the higher fluorescence intensity of the RGE-exosome-treated mice. Small and big red circles indicate tumors and other tissues (liver and spleen), respectively. (**b**) The ROI fluorescence bar graph shows very high fluorescence in the tumor region of RGE-Exo-treated mice due to better accumulation, compared to little fluorescence in free curcumin and DiI-treated mice. Bar represents the mean ± SD (*n* = 6). ★ *p* < 0.05 ★★ *p* < 0.01 [201]. Copyright© 2018. Elsevier Ltd., Amsterdam, The Netherlands.

**Table 1 cells-13-01956-t001:** Clinical trials of MSC-exosomes in various diseases.

No.	Clinical Trial ID	Source of MSC	Phase	Disease	Route of Administration	Status/Outcome
1	NCT03608631	Not Known	I	Pancreatic Cancer	Not known	Trial ongoing
2	NCT06245746	Umbilical cord	I	Acute myeloid leukemia	Intravenous	Trail ongoing
3	NCT03384433	Bone marrow	I/II	Ischemic stroke	Stereotaxis/intra-parenchymal	Trial ongoing
4	NCT00875654	Bone marrow		Ischemic stroke	Intravenous injection	Improved motor function
5	NCT01739777	Umbilical cord	I/II	Heart failure	Intravenous Infusion	improved left ventricular function
6	NCT04388982	Adipose tissue	I/II	Alzheimer’s disease (AD)	Nasal drip	Trial ongoing
7	NCT03172117	Umbilical cord	I	AD	Intracerebro-ventricular injections	Not available
8	NCT05871463	Not Known	IIa	Liver Cirrhosis	Not Known	Not available
9	NCT02138331	Umbilical cord blood	II/III	Type I diabetes mellitus	Intravenous	Trial ongoing
10	NCT02351011	Bone marrow	I/IIa	Osteoarthritis	Intra-articular injection	Improved osteoarthritis symptoms
11	NCT05813379	Not Known	I/II	Skin Rejuvenation	Injection	Not available
12	NCT05078385	Bone marrow	I	Burn wounds	topically applied to the wound	Not available
13	NCT04270006	adipose tissue	I	Periodontitis	Not Known	Not available
Data are compiled from https://clinicaltrials.gov/						

## Data Availability

No new data were created or analyzed in this study. Data sharing is not applicable to this article.

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
