# Peer review of "The Opportunities and Challenges of Mesenchymal Stem Cells-Derived Exosomes in Theranostics and Regenerative Medicine"

_cells, 2024, doi:10.3390/cells13231956_

Round 1
Reviewer 1 Report
Comments and Suggestions for Authors
Review on manuscript entitled
The Opportunities and Challenges of Exosomes in Theranostics and Regenerative Medicine
The present manuscript focuses on the exosomes used in diagnosis and therapy. The authors have processed a large number of articles, including the isolation and characterisation of exosomes. Nevertheless, some questions have been raised.
1. Already several review articles have been published on the diagnostic and therapeutic role of exosomes, focusing on different diseases. This are rather similar articles missing new perspectives. Probably they could be somehow linked.
2. Exosomes and microvesicles co-occur and are referred to as extracellular vesicles (EVs) in conventional terminology. This should be addressed.
3. I do not understand chapter 2.1. It's more about diagnostics, isn't it?
4. The current structure of the main text is difficult to follow. As far as I saw, mostly EVs of stem cell origin are concerned. I suggest putting the emphasis on that.
5. EVs (exosomes) can be used as inherent therapeutics or as drug carriers. This should be concerned.
6. I suggest to divide and discus separately (perhaps in new paper) the role of exosomes in regeneration.
7. The challenges and the outlook are appropriate.
I suggest some re-structuration and revision of the main text for better understanding.
Author Response
1. Already several review articles have been published on the diagnostic and therapeutic role of exosomes, focusing on different diseases. This are rather similar articles missing new perspectives. Probably they could be somehow linked.
Answer: As per the esteemed reviewer’s suggestion, the authors attempted to link the published reviews on this topic (last paragraph of Section 1).
2. Exosomes and microvesicles co-occur and are referred to as extracellular vesicles (EVs) in conventional terminology. This should be addressed.
Answer: The same has been incorporated in the first line of the introduction (section 1) itself.
3. I do not understand chapter 2.1. It's more about diagnostics, isn't it?
Answer: Section 2.1 mainly discusses the inherent therapeutic role of exosomes. Only a few studies have identified specific exosomal RNAs responsible for their particular therapeutic effects, which are referred. The diagnostic role of exosomal RNAs is specifically covered in the theranostic part (Section 2.3).
4. The current structure of the main text is difficult to follow. As far as I saw, mostly EVs of stem cell origin are concerned. I suggest putting the emphasis on that.
Answer: The reviewer is correct, as the present work mainly focuses on MSC-derived exosomes. The authors made necessary changes in the title and abstract to clarify the topic.
5. EVs (exosomes) can be used as inherent therapeutics or as drug carriers. This should be concerned.
Answer: The authors discussed the therapeutic role of exosomes (Section 2.1) in two parts: - a) Inherent therapeutic effects and b) Therapeutic nanocarriers.
The first part emphasized the exosome’s role in treating cancer, cardiovascular, neurological, pancreatic, and hepatic disorders, where specific miRNAs are responsible for such therapeutic effects. The second part is more concerned about their carrier function, such as delivery of pre-loaded drugs, proteins, nucleic acid, etc. Since the discussion is not limited to only drug delivery, we deliberately named the sub-subtitle as ‘therapeutic nanocarriers,’ not simply as ‘drug carriers’.
6. I suggest to divide and discus separately (perhaps in new paper) the role of exosomes in regeneration.
Answer: Cell-based therapy aiming at regeneration is an advanced therapeutic modality for wound healing. Given the drawbacks of cell-based therapy, exosomes can imitate most cell functions and serve as a better cell substitute.
Considering their backgrounds in regenerative medicine, the authors critically examined the regenerative role of exosomes and discussed it in a separate section, without much elaboration of the therapeutic section. They would be pleased if the reviewer kindly accepts this without suggesting for deletion.
Reviewer 2 Report
Comments and Suggestions for Authors
This review highlights the extensive applications of exosomes in therapeutics, diagnosis, and tissue regeneration. Beginning with an overview of exosome properties, biogenesis, isolation, and functional roles, the review then explored recent studies on the therapeutic and regenerative applications of exosomes, with an emphasis on bone, cartilage, periodontal, cardiovascular, skin, and nerve regeneration. It concludes by discussing the theranostic potential of exosomes, followed by an analysis of key challenges, a summary, and future outlook. While the authors have comprehensively summarized recent advances in critical fields with a focus on theranostic applications, certain aspects are underexplored, particularly the administration methods for EV-based therapeutics and the clinical translation of EVs. Thus, a major revision is recommended.
Specific comments are as follows:
1. As MSC-derived exosomes are the focus of this review, please emphasize this EV category in both the abstract and title.
2. Given the potential for oncogenic cargo within EVs, are there concerns regarding the safety of MSC-EVs in cancer treatment? Please discuss any potential risks or adverse effects.
3. Page 7, please make the text formatting consistent.
4. In this review, what kind of EV the authors focused more? Native EVs or bioengineered EV?
5. The authors are recommended to expand the discussion on EV administration, such as aerosol inhalation (e.g., doi.org/10.1016/j.vesic.2022.100002) and scaffold loading (e.g., doi.org/10.1002/advs.202302622). Also targeting systems based on the MSC-EV need more discussion.
6. What about the progress in current MSC EV-related clinical trials? The authors are encouraged to use tables for summarize these trials.
7. Please include a brief overview of exosome-related standards from the EV community, such as those from ISEV (e.g., MISEV 2023, 10.1002/jev2.12404).
8. Please refer to recent literature to summarize critical aspects of EV clinical translation and highlight challenges in manufacturing, isolation, purification, storage, as well as regulatory compliance, which are lacking in the last session of this review. For example: 10.1016/j.tibtech.2024.08.007; doi.org/10.1038/s41565-021-00931-2
Author Response
1. As MSC-derived exosomes are the focus of this review, please emphasize this EV category in both the abstract and title.
Answer: The authors thank the reviewer for this valuable suggestion. The authors have made the necessary changes to the title and abstract.
2. Given the potential for oncogenic cargo within EVs, are there concerns regarding the safety of MSC-EVs in cancer treatment? Please discuss any potential risks or adverse effects.
Answer: The authors are thankful for pointing out this critical issue. The last paragraph of section 3 has been added to address the same.
3. Page 7, please make the text formatting consistent.
Answer: The authors deeply regret this issue, which has been duly resolved.
4. In this review, what kind of EV the authors focused more? Native EVs or bioengineered EV?
Answer: The review covered both native and bioengineered exosomes with modified exosome surface or payload; both essentialy have an MSC origin.
5. The authors are recommended to expand the discussion on EV administration, such as aerosol inhalation (e.g., doi.org/10.1016/j.vesic.2022.100002) and scaffold loading (e.g., doi.org/10.1002/advs.202302622). Also targeting systems based on the MSC-EV need more discussion.
Answer: Based on the suggestion, the other modes of exosome administration, such as inhalation, scaffolds, etc., and the associated targeting systems, have been incorporated in section 1 (introduction, second paragraph).
6. What about the progress in current MSC EV-related clinical trials? The authors are encouraged to use tables for summarize these trials.
Answer: Based on the suggestion, Table 1 has been added to summarize the completed and ongoing clinical trials involving MSC-exosomes in the present context of this review.
7. Please include a brief overview of exosome-related standards from the EV community, such as those from ISEV (e.g., MISEV 2023, 10.1002/jev2.12404).
Answer: A brief overview of exosome-related standards has been provided in Section 3 based on the reviewer's suggestion.
8. Please refer to recent literature to summarize critical aspects of EV clinical translation and highlight challenges in manufacturing, isolation, purification, storage, as well as regulatory compliance, which are lacking in the last session of this review. For example: 10.1016/j.tibtech.2024.08.007; doi.org/10.1038/s41565-021-00931-2
Answer: Thank you for this suggestion. The authors incorporated the suggested contents in Section 3.
Round 2
Reviewer 2 Report
Comments and Suggestions for Authors
The authors have satisfactorily addressed all my concerns.